# Bayesian Dark Knowledge

**Anoop Korattikara, Vivek Rathod, Kevin Murphy**
Google Research
{kbanoop, rathodv, kpmurphy}@google.com

**Max Welling**
University of Amsterdam
m.welling@uva.nl

## Abstract

We consider the problem of Bayesian parameter estimation for deep neural networks, which is important in problem settings where we may have little data, and/or where we need accurate posterior predictive densities $p(y|x, \mathcal{D})$, e.g., for applications involving bandits or active learning. One simple approach to this is to use online Monte Carlo methods, such as SGLD (stochastic gradient Langevin dynamics). Unfortunately, such a method needs to store many copies of the parameters (which wastes memory), and needs to make predictions using many versions of the model (which wastes time).

We describe a method for "distilling" a Monte Carlo approximation to the posterior predictive density into a more compact form, namely a single deep neural network. We compare to two very recent approaches to Bayesian neural networks, namely an approach based on expectation propagation [HLA15] and an approach based on variational Bayes [BCKW15]. Our method performs better than both of these, is much simpler to implement, and uses less computation at test time.

## 1 Introduction

Deep neural networks (DNNs) have recently been achieving state of the art results in many fields. However, their predictions are often over confident, which is a problem in applications such as active learning, reinforcement learning (including bandits), and classifier fusion, which all rely on good estimates of uncertainty.

A principled way to tackle this problem is to use Bayesian inference. Specifically, we first compute the posterior distribution over the model parameters, $p(\theta|\mathcal{D}_N) \propto p(\theta) \prod_{i=1}^{N} p(y_i|x_i, \theta)$, where $\mathcal{D}_N = \{(x_i, y_i)\}_{i=1}^{N}$, $x_i \in \mathcal{X}^D$ is the $i$'th input (where $D$ is the number of features), and $y_i \in \mathcal{Y}$ is the $i$'th output. Then we compute the posterior predictive distribution, $p(y|x, \mathcal{D}_N) = \int p(y|x, \theta)p(\theta|\mathcal{D}_N)d\theta$, for each test point $x$.

For reasons of computational speed, it is common to approximate the posterior distribution by a point estimate such as the MAP estimate, $\hat{\theta} = \operatorname{argmax} p(\theta|\mathcal{D}_N)$. When $N$ is large, we often use stochastic gradient descent (SGD) to compute $\hat{\theta}$. Finally, we make a plug-in approximation to the predictive distribution: $p(y|x, \mathcal{D}_N) \approx p(y|x, \hat{\theta})$. Unfortunately, this loses most of the benefits of the Bayesian approach, since uncertainty in the parameters (which induces uncertainty in the predictions) is ignored.

Various ways of more accurately approximating $p(\theta|\mathcal{D}_N)$ (and hence $p(y|x, \mathcal{D}_N)$) have been developed. Recently, [HLA15] proposed a method called "probabilistic backpropagation" (PBP) based on an online version of expectation propagation (EP), (i.e., using repeated assumed density filtering (ADF)), where the posterior is approximated as a product of univariate Gaussians, one per parameter: $p(\theta|\mathcal{D}_N) \approx q(\theta) \triangleq \prod_i \mathcal{N}(\theta_i|m_i, v_i)$.

An alternative to EP is variational Bayes (VB) where we optimize a lower bound on the marginal likelihood. [Gra11] presented a (biased) Monte Carlo estimate of this lower bound and applies

his method, called "variational inference" (VI), to infer the neural network weights. More recently, [BCKW15] proposed an approach called "Bayes by Backprop" (BBB), which extends the VI method with an unbiased MC estimate of the lower bound based on the "reparameterization trick" of [KW14, RMW14]. In both [Gra11] and [BCKW15], the posterior is approximated by a product of univariate Gaussians.

Although EP and VB scale well with data size (since they use online learning), there are several problems with these methods: (1) they can give poor approximations when the posterior $p(\theta|\mathcal{D}_N)$ does not factorize, or if it has multi-modality or skew; (2) at test time, computing the predictive density $p(y|x, \mathcal{D}_N)$ can be much slower than using the plug-in approximation, because of the need to integrate out the parameters; (3) they need to use double the memory of a standard plug-in method (to store the mean and variance of each parameter), which can be problematic in memory-limited settings such as mobile phones; (4) they can be quite complicated to derive and implement.

A common alternative to EP and VB is to use MCMC methods to approximate $p(\theta|\mathcal{D}_N)$. Traditional MCMC methods are batch algorithms, that scale poorly with dataset size. However, recently a method called stochastic gradient Langevin dynamics (SGLD) [WT11] has been devised that can draw samples approximately from the posterior in an online fashion, just as SGD updates a point estimate of the parameters online. Furthermore, various extensions of SGLD have been proposed, including stochastic gradient hybrid Monte Carlo (SGHMC) [CFG14], stochastic gradient Nosé-Hoover Thermostat (SG-NHT) [DFB$^+$14] (which improves upon SGHMC), stochastic gradient Fisher scoring (SGFS) [AKW12] (which uses second order information), stochastic gradient Riemannian Langevin Dynamics [PT13], distributed SGLD [ASW14], etc. However, in this paper, we will just use "vanilla" SGLD [WT11].[1]

All these MCMC methods (whether batch or online) produce a Monte Carlo approximation to the posterior, $q(\theta) = \frac{1}{S} \sum_{s=1}^{S} \delta(\theta - \theta^s)$, where $S$ is the number of samples. Such an approximation can be more accurate than that produced by EP or VB, and the method is much easier to implement (for SGLD, you essentially just add Gaussian noise to your SGD updates). However, at test time, things are $S$ times slower than using a plug-in estimate, since we need to compute $q(y|x) = \frac{1}{S} \sum_{s=1}^{S} p(y|x, \theta^s)$, and the memory requirements are $S$ times bigger, since we need to store the $\theta^s$. (For our largest experiment, our DNN has 500k parameters, so we can only afford to store a single sample.)

In this paper, we propose to train a parametric model $\mathcal{S}(y|x, w)$ to approximate the Monte Carlo posterior predictive distribution $q(y|x)$ in order to gain the benefits of the Bayesian approach while only using the same run time cost as the plugin method. Following [HVD14], we call $q(y|x)$ the "teacher" and $\mathcal{S}(y|x, w)$ the "student". We use SGLD[2] to estimate $q(\theta)$ and hence $q(y|x)$ online; we simultaneously train the student online to minimize $\mathrm{KL}(q(y|x)||\mathcal{S}(y|x, w))$. We give the details in Section 2.

Similar ideas have been proposed in the past. In particular, [SG05] also trained a parametric student model to approximate a Monte Carlo teacher. However, they used batch training and they used mixture models for the student. By contrast, we use online training (and can thus handle larger datasets), and use deep neural networks for the student.

[HVD14] also trained a student neural network to emulate the predictions of a (larger) teacher network (a process they call "distillation"), extending earlier work of [BCNM06] which approximated an ensemble of classifiers by a single one. The key difference from our work is that our teacher is generated using MCMC, and our goal is not just to improve classification accuracy, but also to get reliable probabilistic predictions, especially away from the training data. [HVD14] coined the term "dark knowledge" to represent the information which is "hidden" inside the teacher network, and which can then be distilled into the student. We therefore call our approach "Bayesian dark knowledge".

In summary, our contributions are as follows. First, we show how to combine online MCMC methods with model distillation in order to get a simple, scalable approach to Bayesian inference of the parameters of neural networks (and other kinds of models). Second, we show that our probabilistic predictions lead to improved log likelihood scores on the test set compared to SGD and the recently proposed EP and VB approaches.

## 2 Methods

Our goal is to train a student neural network (SNN) to approximate the Bayesian predictive distribution of the teacher, which is a Monte Carlo ensemble of teacher neural networks (TNN).

If we denote the predictions of the teacher by $p(y|x, \mathcal{D}_N)$ and the parameters of the student network by $w$, our objective becomes

$$L(w|x) = \mathrm{KL}(p(y|x, \mathcal{D}_N)||\mathcal{S}(y|x, w)) = -\mathbb{E}_{p(y|x,\mathcal{D}_N)} \log \mathcal{S}(y|x, w) + \mathrm{const}$$

$$= -\int \left[ \int p(y|x, \theta) p(\theta|D_N) d\theta \right] \log \mathcal{S}(y|x, w) dy$$

$$= -\int p(\theta|D_N) \int p(y|x, \theta) \log \mathcal{S}(y|x, w) dy \ d\theta$$

$$= -\int p(\theta|D_N) \left[ \mathbb{E}_{p(y|x,\theta)} \log \mathcal{S}(y|x, w) \right] d\theta \tag{1}$$

Unfortunately, computing this integral is not analytically tractable. However, we can approximate this by Monte Carlo:

$$\hat{L}(w|x) = -\frac{1}{|\Theta|} \sum_{\theta^s \in \Theta} \mathbb{E}_{p(y|x,\theta^s)} \log \mathcal{S}(y|x, w) \tag{2}$$

where $\Theta$ is a set of samples from $p(\theta|\mathcal{D}_N)$.

To make this a function just of $w$, we need to integrate out $x$. For this, we need a dataset to train the student network on, which we will denote by $\mathcal{D}'$. Note that points in this dataset do not need ground truth labels; instead the labels (which will be probability distributions) will be provided by the teacher. The choice of student data controls the domain over which the student will make accurate predictions. For low dimensional problems (such as in Section 3.1), we can uniformly sample the input domain. For higher dimensional problems, we can sample "near" the training data, for example by perturbing the inputs slightly. In any case, we will compute a Monte Carlo approximation to the loss as follows:

$$\hat{L}(w) = \int p(x) L(w|x) dx \approx \frac{1}{|\mathcal{D}'|} \sum_{x' \in \mathcal{D}'} L(w|x')$$

$$\approx -\frac{1}{|\Theta|} \frac{1}{|\mathcal{D}'|} \sum_{\theta^s \in \Theta} \sum_{x' \in \mathcal{D}'} \mathbb{E}_{p(y|x',\theta^s)} \log \mathcal{S}(y|x', w) \tag{3}$$

It can take a lot of memory to pre-compute and store the set of parameter samples $\Theta$ and the set of data samples $\mathcal{D}'$, so in practice we use the stochastic algorithm shown in Algorithm 1, which uses a single posterior sample $\theta^s$ and a minibatch of $x'$ at each step.

The hyper-parameters $\lambda$ and $\gamma$ from Algorithm 1 control the strength of the priors for the teacher and student networks. We use simple spherical Gaussian priors (equivalent to $L_2$ regularization); we set the precision (strength) of these Gaussian priors by cross-validation. Typically $\lambda \gg \gamma$, since the student gets to "see" more data than the teacher. This is true for two reasons: first, the teacher is trained to predict a single label per input, whereas the student is trained to predict a distribution, which contains more information (as argued in [HVD14]); second, the teacher makes multiple passes over the same training data, whereas the student sees "fresh" randomly generated data $\mathcal{D}'$ at each step.

### 2.1 Classification

For classification problems, each teacher network $\theta^s$ models the observations using a standard softmax model, $p(y = k|x, \theta^s)$. We want to approximate this using a student network, which also has a

---

**Algorithm 1:** Distilled SGLD

---

Input: $\mathcal{D}_N = \{(x_i, y_i)\}_{i=1}^N$, minibatch size $M$, number of iterations $T$, teacher learning schedule $\eta_t$, student learning schedule $\rho_t$, teacher prior $\lambda$, student prior $\gamma$

**for** $t = 1 : T$ **do**

   // Train teacher (SGLD step)

   Sample minibatch indices $S \subset [1, N]$ of size $M$

   Sample $z_t \sim \mathcal{N}(0, \eta_t I)$

   Update $\theta_{t+1} := \theta_t + \frac{\eta_t}{2} \left( \nabla_\theta \log p(\theta | \lambda) + \frac{N}{M} \sum_{i \in S} \nabla_\theta \log p(y_i | x_i, \theta) \right) + z_t$

   // Train student (SGD step)

   Sample $\mathcal{D}'$ of size $M$ from student data generator

   $w_{t+1} := w_t - \rho_t \left( \frac{1}{M} \sum_{x' \in \mathcal{D}'} \nabla_w \hat{L}(w, \theta_{t+1} | x') + \gamma w_t \right)$

---

softmax output, $\mathcal{S}(y = k | x, w)$. Hence from Eqn. 2, our loss function estimate is the standard cross entropy loss:

$$\hat{L}(w | \theta^s, x) = -\sum_{k=1}^K p(y = k | x, \theta^s) \log \mathcal{S}(y = k | x, w) \tag{4}$$

The student network outputs $\beta_k(x, w) = \log \mathcal{S}(y = k | x, w)$. To estimate the gradient w.r.t. $w$, we just have to compute the gradients w.r.t. $\beta$ and back-propagate through the network. These gradients are given by $\frac{\partial \hat{L}(w, \theta^s | x)}{\partial \beta_k(x, w)} = -p(y = k | x, \theta^s)$.

### 2.2 Regression

In regression, the observations are modeled as $p(y_i | x_i, \theta) = \mathcal{N}(y_i | f(x_i | \theta), \lambda_n^{-1})$ where $f(x | \theta)$ is the prediction of the TNN and $\lambda_n$ is the noise precision. We want to approximate the predictive distribution as $p(y | x, \mathcal{D}_N) \approx \mathcal{S}(y | x, w) = \mathcal{N}(y | \mu(x, w), e^{\alpha(x,w)})$. We will train a student network to output the parameters of the approximating distribution $\mu(x, w)$ and $\alpha(x, w)$; note that this is twice the number of outputs of the teacher network, since we want to capture the (data dependent) variance.[3] We use $e^{\alpha(x,w)}$ instead of directly predicting the variance $\sigma^2(x | w)$ to avoid dealing with positivity constraints during training.

To train the SNN, we will minimize the objective defined in Eqn. 2:

$$
\begin{aligned}
\hat{L}(w | \theta^s, x) &= -\mathbb{E}_{p(y | x, \theta^s)} \log \mathcal{N}(y | \mu(x, w), e^{\alpha(x,w)}) \\
&= \frac{1}{2} \mathbb{E}_{p(y | x, \theta^s)} \left[ \alpha(x, w) + e^{-\alpha(x,w)} (y - \mu(x, w)^2) \right] \\
&= \frac{1}{2} \left[ \alpha(x, w) + e^{-\alpha(x,w)} \left\{ (f(x | \theta^s) - \mu(x, w))^2 + \frac{1}{\lambda_n} \right\} \right]
\end{aligned}
$$

Now, to estimate $\nabla_w \hat{L}(w, \theta^s | x)$, we just have to compute $\frac{\partial \hat{L}}{\partial \mu(x, w)}$ and $\frac{\partial \hat{L}}{\partial \alpha(x, w)}$, and back propagate through the network. These gradients are:

$$\frac{\partial \hat{L}(w, \theta^s | x)}{\partial \mu(x, w)} = e^{-\alpha(x,w)} \{ \mu(x, w) - f(x | \theta^s) \} \tag{5}$$

$$\frac{\partial \hat{L}(w, \theta^s | x}{\partial \alpha(x, w)} = \frac{1}{2} \left[ 1 - e^{-\alpha(x,w)} \left\{ (f(x | \theta^s) - \mu(x, w))^2 + \frac{1}{\lambda_n} \right\} \right] \tag{6}$$

## 3 Experimental results

In this section, we compare SGLD and distilled SGLD with other approximate inference methods, including the plugin approximation using SGD, the PBP approach of [HLA15], the BBB approach of

| Dataset | $N$ | $D$ | $\mathcal{Y}$ | PBP | BBB | HMC |
|---|---|---|---|---|---|---|
| ToyClass | 20 | 2 | $\{0,1\}$ | N | N | Y |
| MNIST | 60k | 784 | $\{0,\dots,9\}$ | N | Y | N |
| ToyReg | 10 | 1 | $\mathbb{R}$ | Y | N | Y |
| Boston Housing | 506 | 13 | $\mathbb{R}$ | Y | N | N |

Table 1: Summary of our experimental configurations.

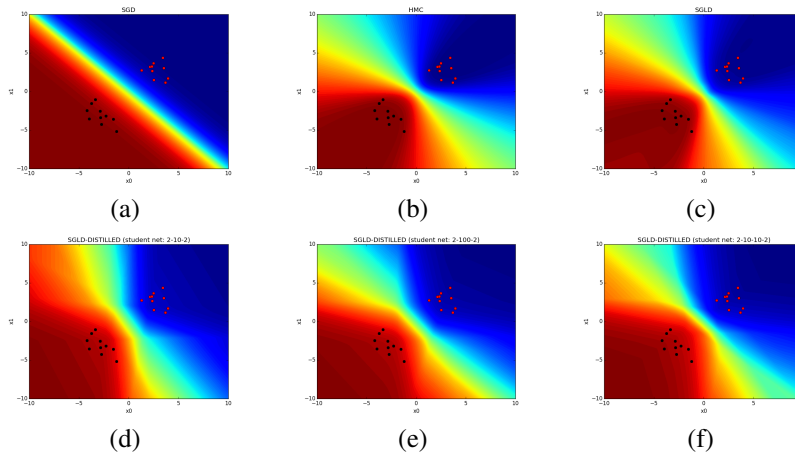

Figure 1: Posterior predictive density for various methods on the toy 2d dataset. (a) SGD (plugin) using the 2-10-2 network. (b) HMC using 20k samples. (c) SGLD using 1k samples. (d-f) Distilled SGLD using a student network with the following architectures: 2-10-2, 2-100-2 and 2-10-10-2.

[BCKW15], and Hamiltonian Monte Carlo (HMC) [Nea11], which is considered the "gold standard" for MCMC for neural nets. We implemented SGD and SGLD using the Torch library (`torch.ch`). For HMC, we used Stan (`mc-stan.org`). We perform this comparison for various classification and regression problems, as summarized in Table 1.[4]

## 3.1 Toy 2d classification problem

We start with a toy 2d binary classification problem, in order to visually illustrate the performance of different methods. We generate a synthetic dataset in 2 dimensions with 2 classes, 10 points per class. We then fit a multi layer perceptron (MLP) with one hidden layer of 10 ReLu units and 2 softmax outputs (denoted 2-10-2) using SGD. The resulting predictions are shown in Figure 1(a). We see the expected sigmoidal probability ramp orthogonal to the linear decision boundary. Unfortunately, this method predicts a label of 0 or 1 with very high confidence, even for points that are far from the training data (e.g., in the top left and bottom right corners).

In Figure 1(b), we show the result of HMC using 20k samples. This is the "true" posterior predictive density which we wish to approximate. In Figure 1(c), we show the result of SGLD using about 1000 samples. Specifically, we generate 100k samples, discard the first 2k for burnin, and then keep every 100'th sample. We see that this is a good approximation to the HMC distribution.

In Figures 1(d-f), we show the results of approximating the SGLD Monte Carlo predictive distribution with a single student MLP of various sizes. To train this student network, we sampled points at random from the domain of the input, $[-10, 10] \times [-10, 10]$; this encourages the student to predict accurately at all locations, including those far from the training data. In (d), the student has the same

| Model | Num. params. | KL |
|---|---|---|
| SGD | 40 | 0.246 |
| SGLD | 40k | 0.007 |
| Distilled 2-10-2 | 40 | 0.031 |
| Distilled 2-100-2 | 400 | 0.014 |
| Distilled 2-10-10-2 | 140 | 0.009 |

Table 2: KL divergence on the 2d classification dataset.

| SGD [BCKW15] | Dropout | BBB | SGD (our impl.) | SGLD | Dist. SGLD |
|---|---|---|---|---|---|
| 1.83 | 1.51 | 1.82 | $1.536 \pm 0.0120$ | $1.271 \pm 0.0126$ | $1.307 \pm 0.0169$ |

Table 3: Test set misclassification rate on MNIST for different methods using a 784-400-400-10 MLP. SGD (first column), Dropout and BBB numbers are quoted from [BCKW15]. For our implmentation of SGD (fourth column), SGLD and distilled SGLD, we report the mean misclassification rate over 10 runs and its standard error.

size as the teacher (2-10-2), but this is too simple a model to capture the complexity of the predictive distribution (which is an average over models). In (e), the student has a larger hidden layer (2-100-2); this works better. However, we get best results using a two hidden layer model (2-10-10-2), as shown in (f).

In Table 2, we show the KL divergence between the HMC distribution (which we consider as ground truth) and the various approximations mentioned above. We computed this by comparing the probability distributions pointwise on a 2d grid. The numbers match the qualitative results shown in Figure 1.

## 3.2 MNIST classification

Now we consider the MNIST digit classification problem, which has $N = 60k$ examples, 10 classes, and $D = 784$ features. The only preprocessing we do is divide the pixel values by 126 (as in [BCKW15]). We train only on 50K datapoints and use the remaining 10K for tuning hyperparameters. This means our results are not strictly comparable to a lot of published work, which uses the whole dataset for training; however, the difference is likely to be small.

Following [BCKW15], we use an MLP with 2 hidden layers with 400 hidden units per layer, ReLU activations, and softmax outputs; we denote this by 784-400-400-10. This model has 500k parameters.

We first fit this model by SGD, using these hyper parameters: fixed learning rate of $\eta_t = 5 \times 10^{-6}$, prior precision $\lambda = 1$, minibatch size $M = 100$, number of iterations $T = 1M$. As shown in Table 3, our final error rate on the test set is 1.536%, which is a bit lower than the SGD number reported in [BCKW15], perhaps due to the slightly different training/ validation configuration.

Next we fit this model by SGLD, using these hyper parameters: fixed learning rate of $\eta_t = 4 \times 10^{-6}$, thinning interval $\tau = 100$, burn in iterations $B = 1000$, prior precision $\lambda = 1$, minibatch size $M = 100$. As shown in Table 3, our final error rate on the test set is about 1.271%, which is better than the SGD, dropout and BBB results from [BCKW15].[5]

Finally, we consider using distillation, where the teacher is an SGLD MC approximation of the posterior predictive. We use the same 784-400-400-10 architecture for the student as well as the teacher. We generate data for the student by adding Gaussian noise (with standard deviation of 0.001) to randomly sampled training points[6] We use a constant learning rate of $\rho = 0.005$, a batch size of $M = 100$, a prior precision of 0.001 (for the student) and train for $T = 1M$ iterations. We obtain a test error of 1.307% which is very close to that obtained with SGLD (see Table 4).

| SGD | SGLD | Distilled SGLD |
|---|---|---|
| -0.0613 ± 0.0002 | -0.0419 ± 0.0002 | -0.0502 ± 0.0007 |

Table 4: Log likelihood per test example on MNIST. We report the mean over 10 trials ± one standard error.

| Method | Avg. test log likelihood |
|---|---|
| PBP (as reported in [HLA15]) | -2.574 ± 0.089 |
| VI (as reported in [HLA15]) | -2.903 ± 0.071 |
| SGD | -2.7639 ± 0.1527 |
| SGLD | -2.306 ± 0.1205 |
| SGLD distilled | -2.350 ± 0.0762 |

Table 5: Log likelihood per test example on the Boston housing dataset. We report the mean over 20 trials ± one standard error.

We also report the average test log-likelihood of SGD, SGLD and distilled SGLD in Table 4. The log-likelihood is equivalent to the *logarithmic scoring rule* [Bic07] used in assessing the calibration of probabilistic models. The logarithmic rule is a strictly proper scoring rule, meaning that the score is uniquely maximized by predicting the true probabilities. From Table 4, we see that both SGLD and distilled SGLD acheive higher scores than SGD, and therefore produce better calibrated predictions.

Note that the SGLD results were obtained by averaging predictions from $\approx 10,000$ models sampled from the posterior, whereas distillation produces a single neural network that approximates the average prediction of these models, i.e. distillation reduces both storage and test time costs of SGLD by a factor of 10,000, without sacrificing much accuracy. In terms of training time, SGD took 1.3 ms, SGLD took 1.6 ms and distilled SGLD took 3.2 ms per iteration. In terms of memory, distilled SGLD requires only twice as much as SGD or SGLD during training, and the same as SGD during testing.

## 3.3 Toy 1d regression

We start with a toy 1d regression problem, in order to visually illustrate the performance of different methods. We use the same data and model as [HLA15]. In particular, we use $N = 20$ points in $D = 1$ dimensions, sampled from the function $y = x^3 + \epsilon_n$, where $\epsilon_n \sim \mathcal{N}(0, 9)$. We fit this data with an MLP with 10 hidden units and ReLU activations. For SGLD, we use $S = 2000$ samples. For distillation, the teacher uses the same architecture as the student.

The results are shown in Figure 2. We see that SGLD is a better approximation to the "true" (HMC) posterior predictive density than the plugin SGD approximation (which has no predictive uncertainty), and the VI approximation of [Gra11]. Finally, we see that distilling SGLD incurs little loss in accuracy, but saves a lot computationally.

## 3.4 Boston housing

Finally, we consider a larger regression problem, namely the Boston housing dataset, which was also used in [HLA15]. This has $N = 506$ data points (456 training, 50 testing), with $D = 13$ dimensions. Since this data set is so small, we repeated all experiments 20 times, using different train/ test splits.

Following [HLA15], we use an MLP with 1 layer of 50 hidden units and ReLU activations. First we use SGD, with these hyper parameters[7]: Minibatch size $M = 1$, noise precision $\lambda_n = 1.25$, prior precision $\lambda = 1$, number of trials 20, constant learning rate $\eta_t = 1e - 6$, number of iterations $T = 170K$. As shown in Table 5, we get an average log likelihood of $-2.7639$.

Next we fit the model using SGLD. We use an initial learning rate of $\eta_0 = 1e - 5$, which we reduce by a factor of 0.5 every 80K iterations; we use 500K iterations, a burnin of 10K, and a thinning

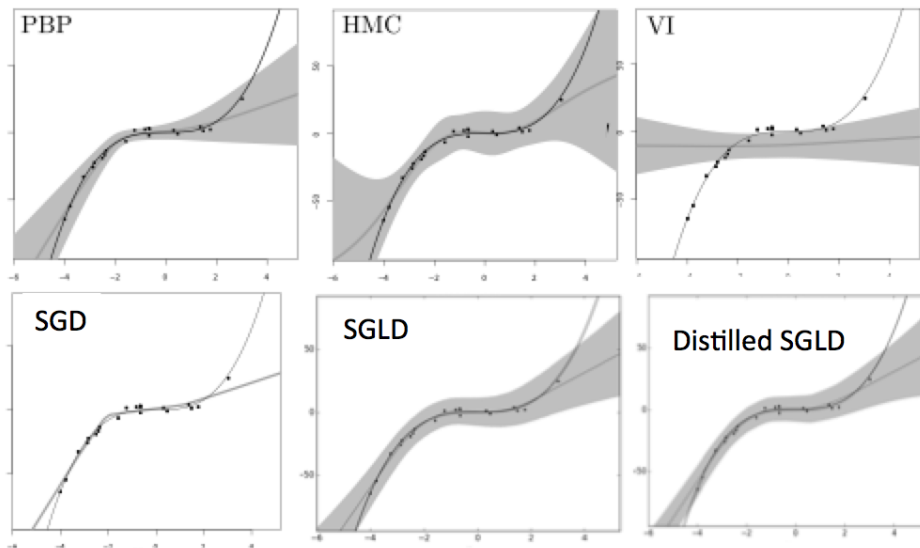

Figure 2: Predictive distribution for different methods on a toy 1d regression problem. (a) PBP of [HLA15]. (b) HMC. (c) VI method of [Gra11]. (d) SGD. (e) SGLD. (f) Distilled SGLD. Error bars denote 3 standard deviations. (Figures a-d kindly provided by the authors of [HLA15]. We replace their term "BP" (backprop) with "SGD" to avoid confusion.)

interval of 10. As shown in Table 5, we get an average log likelihood of $-2.306$, which is better than SGD.

Finally, we distill our SGLD model. The student architecture is the same as the teacher. We use the following teacher hyper parameters: prior precision $\lambda = 2.5$; initial learning rate of $\eta_0 = 1e-5$, which we reduce by a factor of 0.5 every 80K iterations. For the student, we use generated training data with Gaussian noise with standard deviation 0.05, we use a prior precision of $\gamma = 0.001$, an initial learning rate of $\rho_0 = 1e-2$, which we reduce by 0.8 after every $5e3$ iterations. As shown in Table 5, we get an average log likelihood of $-2.350$, which is only slightly worse than SGLD, and much better than SGD. Furthermore, both SGLD and distilled SGLD are better than the PBP method of [HLA15] and the VI method of [Gra11].

## 4    Conclusions and future work

We have shown a very simple method for "being Bayesian" about neural networks (and other kinds of models), that seems to work better than recently proposed alternatives based on EP [HLA15] and VB [Gra11, BCKW15].

There are various things we would like to do in the future: (1) Show the utility of our model in an end-to-end task, where predictive uncertainty is useful (such as with contextual bandits or active learning). (2) Consider ways to reduce the variance of the algorithm, perhaps by keeping a running minibatch of parameters uniformly sampled from the posterior, which can be done online using reservoir sampling. (3) Exploring more intelligent data generation methods for training the student. (4) Investigating if our method is able to reduce the prevalence of confident false predictions on adversarially generated examples, such as those discussed in [SZS+14].

### Acknowledgements

We thank José Miguel Hernández-Lobato, Julien Cornebise, Jonathan Huang, George Papandreou, Sergio Guadarrama and Nick Johnston.

## Footnotes

[1] We did some preliminary experiments with SG-NHT for fitting an MLP to MNIST data, but the results were not much better than SGLD.

[2] Note that SGLD is an approximate sampling algorithm and introduces a slight bias in the predictions of the teacher and student network. If required, we can replace SGLD with an exact MCMC method (e.g. HMC) to get more accurate results at the expense of more training time.

[3] This is not necessary in the classification case, since the softmax distribution already captures uncertainty.

[4] Ideally, we would apply all methods to all datasets, to enable a proper comparison. Unfortunately, this was not possible, for various reasons. First, the open source code for the EP approach only supports regression, so we could not evaluate this on classification problems. Second, we were not able to run the BBB code, so we just quote performance numbers from their paper [BCKW15]. Third, HMC is too slow to run on large problems, so we just applied it to the small "toy" problems. Nevertheless, our experiments show that our methods compare favorably to these other methods.

[5] We only show the BBB results with the same Gaussian prior that we use. Performance of BBB can be improved using other priors, such as a scale mixture of Gaussians, as shown in [BCKW15]. Our approach could probably also benefit from such a prior, but we did not try this.

[6] In the future, we would like to consider more sophisticated data perturbations, such as elastic distortions.

[7]We choose all hyper-parameters using cross-validation whereas [HLA15] performs posterior inference on the noise and prior precisions, and uses Bayesian optimization to choose the remaining hyper-parameters.

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
