[Reviews · NeurIPS 2015]

Submitted by Assigned_Reviewer_1

The proposed algorithm uses the KL distance between the true and the proposed posterior as the objective function. To compute the KL distance it uses stochastic gradient Langevin dynamics to effienctly sample from the posterior and to optimize it it uses stochastic gradient descent.

The writing is quite clear. It is a simple but effective approach and the examples provided show its benefits quite convincingly.

Summary: The paper proposes a stochastic gradient algorithm to train a parametric model of the posterior predictive distribution of a Bayesian model.

Submitted by Assigned_Reviewer_2

the main idea of the manuscript is, in a regression context, to approximate the predictive distribution with a simple tractable model in order to avoid the high cost required to approximate the true predictive distribution using Monte Carlo samples. The authors discuss why they want to use a Bayesian approach (and their claims are somewhat supported by their simulations) in contrast with a plug in approach. The approach developed comes with no surprise, but is natural and sensible : minimize an averaged KL divergence between the true predictive and the surrogate parametric model using the output of SGLD algorithm. Performance of the method is illustrated through simulations on some examples and, as for any manuscript, outperforms competitors. There are numerous parameters to choose in the algorithm, which is not discussed.
Summary: Nothing surprising in the manuscript. It is an idea worth exploring.

Submitted by Assigned_Reviewer_3

The paper uses an online approximation to MCMC to draw parameters for a Bayesian neural network. The predictive distribution under these samples is then fitted using stochastic approximation. The comparisons are to recent work on approximate Bayesian inference applied to the same models and example problems. The claim being that it's a better method (at least in some ways). The paper does not yet present demonstrate that these methods will push forward any particular application.

The paper is a fairly natural extension of existing work. It is to-the-point, done well, presented clearly, and would be easy for others to try, so could have considerable impact.

Some indication of training times would be useful. Presumably the training time is much slower than the other methods, as the abstract makes a point specifically about test time. However, this issue is simply dodged.

I think it should be made clear in the paper that stochastic gradient Langevin dynamics only draws approximate samples from the posterior. Not just in the usual MCMC sense of requiring burn-in, but it doesn't even respect the posterior locally when exploring some mode for a long time. While there are various developments (cited), none of these offer the same sort of results as traditional batch MCMC on small problems. It's true that on large problems there isn't a better option, but I think it's important not to over-promise. The eagle-eyed reader will notice the posterior is wrong in Figure 2, but it's never really pointed out.

I would remove the statement that the priors are equivalent to L_2 regularization. That's only if doing MAP estimation, which this work is emphatically not! It seems a shame to use simple spherical Gaussian priors, given all the work on Bayesian neural networks in the 1990s. It's hard to take these priors too seriously, especially in a very large and deep network.

In an engineering sense, the overall procedure is useful. Although future comparisons would be required to see if it really beats standard regularization, early stopping, drop-out, and various other hacks to avoid overfitting. In applications where the predictive distributions are the goal of inference, more work is also required to make approximate Bayesian inference at this scale trustworthy. However, this is an interesting step in the process.

Minor:

I personally don't think the title really captures the contribution. It may sound cool, but there isn't any discussion of eeking out "hidden" knowledge in the paper.

I don't think the fitting algorithm is really plain SGD. The subsequent theta samples in the final line are dependent, so presumably some stochastic approximation argument needs to be made.

$5e-6$ is ugly typesetting. I suggest $5\!\times\!10^{-6}$, and $1e-5$ could just be $10^{-5}$.

References: The NIPS styleguide says the references should use the numerical "unsrt" style. The references could have more complete details. Some proper nouns need capitalizing. Some authors only have initials, while most names are listed in full.
Summary: This paper is a natural extension of Snelson and Ghahrahmani's "Compact approximations to Bayesian predictive distributions" (ICML 2005), using neural networks and online methods for the MCMC and gradient-based fitting. It's well done, bringing existing neat ideas up to date.

Submitted by Assigned_Reviewer_4

Paper Title: Bayesian Dark Knowledge

Paper Summary: This paper presents a method for approximately learning a Bayesian neural network model while avoiding major storage costs accumulated during training and computational costs during prediction. Typically, in Bayesian models, samples are generated, and a sample approximation to the posterior predictive distribution is formed. However, this requires storing many copies of the parameters of a model, which may require a great deal of storage/memory in high-parameter neural network models. Additionally, prediction on test data requires evaluation of all sampled models, which may be computationally costly. This paper aims to learn a model that makes (approximate) posterior-predictive-based predictions on test data without storing many copies of the parameter set. The method presented here accomplishes this by training a "student" model to approximate a Bayesian "teacher's" predictions---a procedure in the neural networks literature referred to as "distillation" (and also referred to in previous papers as "model compression"). However, the student and teacher are trained simultaneously in an online manner (so no large collection of samples are required to be stored, even during training). Experiments are shown on synthetic data (a toy 2D binary classification problem and a toy 1D regression problem), on classification for the MNIST dataset, and on regression for a housing dataset.

Comments:

- I feel that working to develop computational tools for practical Bayesian inference in neural networks is an important direction of research (in particular, tools that mitigate the issue of large storage/memory requirements when sampling in large-parameters Bayesian models), and I feel that this paper is making good steps in this direction.

- One issue I have with the novelty of this paper is that the fundamental concept being developed here (student/teacher learning, or model distillation/compression) is not new --- the authors apply this existing idea in a new domain. However, I do feel the main novelty (and strength) of this paper is the clever way in which the algorithm carries out simultaneous online training of the teacher and student without requiring storage of samples. This is an online distillation/compression method in which the teacher is never "fully trained" (the teacher never actually provides full posterior-predictive "labels" for the student) and yet the student is still able to learn the "full trained" teacher's model.

- One potential computational issue with the presented algorithm is that the student must be trained (to mimic the teacher), on the training dataset D', at every iteration during the MCMC algorithm. I believe this requires a great deal more training of the student than in typical distillation methods (which do not need to make many passes through the training dataset D'). This might be particularly problematic if D' is required to be large in order to achieve good performance. In general, I feel that there is not enough discussion about the training set D' (whether particular choices about D' affect the method's performance, and the particular details used in the experiments presented in this paper).

- I feel that the experiments in this paper were not totally polished and thorough. The authors write that they could not enable a proper comparison of all methods on all datasets because (in part) "the open source code for the EP approach only supports regression" and "we did not get access to the code for the VB approach in time for us to compare to it". I appreciate the honesty here! However, it would be nice to complete these goals and polish the experiments section in this paper.
Summary: I feel that the goal of this paper---to develop methods for Bayesian neural networks without the need to store and evaluate many copies of the model---is important, and that the presented method is a clever way of approaching this goal (though it is, in part, an application of existing methods). However, the paper could do a good deal more to polish the experiments section.

Author Feedback
Author rebuttal: We thank the reviewers for their helpful comments. Our responses to the main questions raised in the review:

1. "No mention of training times" (Assigned_Reviewer_2)

- The time for one iteration of our algorithm is approximately equal to the time for 1 iteration of SGLD on the teacher plus 1 iteration of SGD on the student. For the MNIST experiment with 400 Hidden Units/layer, our implementation took 1.3 secs for SGD on the teacher, 1.6 secs for SGLD on the teacher, and 3 secs for online distillation, i.e. SGLD on the teacher + SGD on the student (times are given in secs/1000 iterations).

2. "...it should be made clear that SGLD only draws approximate samples... " (Assigned_Reviewer_2)

- Yes, we will make this clear. But please note that, if we want exact posterior samples, we can use a traditional MCMC method such as HMC for inference in the teacher network.

3. "The subsequent theta samples in the final line are dependent, so some stochastic approximation argument needs to be made." (Assigned_Reviewer_2)
"I think SGLD theta samples should be thinned as there will be high auto-correlation" (Assigned_Reviewer_6)

- Yes, high autocorrelation between subsequent samples may in theory slow down the convergence of SGD on the student. We want the SGLD chain to mix faster than the SGD chain converges, which we achieve using a high step size for SGLD and annealing the step size quickly for SGD. Another way to break correlations between subsequent samples is to generate a pool of samples using SGLD ahead of time, and then randomly select from this pool in each iteration of SGD. This might be not be possible in practice with large networks because of storage costs, in which case we can maintain a smaller pool of samples using reservoir sampling. Other options are thinning or getting samples from multiple parallel SGLD chains.

4. "there is not enough discussion about the training set D'" (Assigned_Reviewer_3)

- The purpose of the dataset D' is to provide data outside the training set where we want the student to mimic the teacher well. For our toy classification example, D' is a grid because we wanted to visualize the predictive probability. For real data, D' can be restricted to data that we expect to see in practice. For example, for image data, we can use 1) perturbed versions of the training data e.g. by adding Gaussian noise or by elastic distortions, 2) unlabeled images and/or 3) adversarial examples.

5. "...requires a great deal more training of the student than in typical distillation methods (which do not need to make many passes through the training dataset D')..." (Assigned_Reviewer_3)

- Note that D` can even be an infinite dataset. We use only a mini-batch of data to make each update. Just as SGD can learn from seeing each data point only a few times, our student network can learn to approximate the teacher without too many passes, and usually converges in just a little more time than it takes for SGLD to converge.

6. "I feel that the experiments in this paper were not totally polished and thorough" (Assigned_Reviewer_3)

- We just received code from Blundell et al. and are working on finishing the comparisons with their method. We are also working with astrophysicists to build predictive models for the outcomes of expensive experiments at the Large Hadron Collider. This is a real world example where modeling uncertainty is important. If the paper is accepted, we will add these results to the camera ready version.

7. "The advantage of distilled SGLD over SGLD is not well supported in the experiments." (Assigned_Reviewer_5)

- In terms of accuracy of predictions and uncertainty estimates, distillation has no advantage over SGLD. But SGLD and other MCMC methods have to store many posterior samples which is impractical both in terms of memory/storage costs and test time computational costs. The goal of distilled SGLD is to avoid these computational problems, not to improve accuracy.

8. "I do not like the model fit in line 190... you are losing correlations between pairs (x,w) and (x',w')..." (Assigned_Reviewer_6)

- We are sorry but we did not quite understand this remark. Please note that we are only interested in the predictive distribution p(y|x,Data), and not in the posterior distribution over the weights of the neural network.